

# PhyloPrimer: a taxon-specific oligonucleotide design platform

Gilda Varliero[1], Jared Wray[1], Cédric Malandain[2] and Gary Barker[1]

[1] School of Biological Sciences, University of Bristol, Bristol, UK
[2] Environmental Expertise, HYDREKA, Lyon, France

## ABSTRACT

Many environmental and biomedical biomonitoring and detection studies aim to explore the presence of specific organisms or gene functionalities in microbiome samples. In such cases, when the study hypotheses can be answered with the exploration of a small number of genes, a targeted PCR-approach is appropriate. However, due to the complexity of environmental microbial communities, the design of specific primers is challenging and can lead to non-specific results. We designed PhyloPrimer, the first user-friendly platform to semi-automate the design of taxon-specific oligos (i.e., PCR primers) for a gene of interest. The main strength of PhyloPrimer is the ability to retrieve and align GenBank gene sequences matching the user's input, and to explore their relationships through an online dynamic tree. PhyloPrimer then designs oligos specific to the gene sequences selected from the tree and uses the tree non-selected sequences to look for and maximize oligo differences between targeted and non-targeted sequences, therefore increasing oligo taxon-specificity (positive/negative consensus approach). Designed oligos are then checked for the presence of secondary structure with the nearest-neighbor (NN) calculation and the presence of off-target matches with *in silico* PCR tests, also processing oligos with degenerate bases. Whilst the main function of PhyloPrimer is the design of taxon-specific oligos (down to the species level), the software can also be used for designing oligos to target a gene without any taxonomic specificity, for designing oligos from preselected sequences and for checking predesigned oligos. We validated the pipeline on four commercially available microbial mock communities using PhyloPrimer to design genus- and species-specific primers for the detection of *Streptococcus* species in the mock communities. The software performed well on these mock microbial communities and can be found at https://www.cerealsdb.uk.net/cerealgenomics/phyloprimer.

## INTRODUCTION

The Polymerase Chain Reaction (PCR) is a pivotal technique to many molecular protocols and is widely used to exponentially amplify a specific portion of DNA (e.g., gene) using DNA or RNA template (e.g., the entire DNA or RNA content of an environmental sample), primers, deoxynucleotides (dNTPs), DNA polymerase and reaction buffers (*Garibyan & Avashia, 2013*). Before starting with any PCR-based procedure, primers need to be selected to target the specific DNA region and organisms. The amplification starts where the primers

Corresponding author
Gilda Varliero,
gilda.varliero@bristol.ac.uk

anneal to the DNA template, for this reason the specificity of the PCR reaction is highly impacted by the specificity of the primers to the DNA template.

The design of new oligonucleotides (i.e., primers or probes), hereafter abbreviated as oligos, is a relatively easy task when working with known axenic cultures or known low complexity communities but can be challenging when dealing with unknown organisms and complex environmental communities. Different studies can require different level of oligo-specificity: oligos could be designed to target the same DNA portion in all the community organisms (e.g., universal primers), in a specific group of organisms or in a specific species or strain. The latter two tasks become challenging when the target DNA fragment is present in non-target organisms that are part of the community (*Fierer et al., 2005*).

Many different primer and probe sequences have been published. These oligos can target a broad variety of different DNA sequences and can present a wide range of target organism's specificity. Universal oligos, such as primers targeting housekeeping genes (e.g., 16S rRNA gene) are widely used for the study of microbial diversity and in diagnostic surveys (e.g., *Takahashi et al., 2014*). It is also possible to target non-universal genes, such as the *nifH* gene (e.g., *Gaby & Buckley, 2012*) and the *pmoA* gene (e.g., *Wang et al., 2017*), in order to target only organisms with a specific metabolism and that occupy specific environmental niches. Oligos can also have a more specific target: they can amplify only genes present in organisms of interest even when the gene is present in a wider selection of organisms (e.g., *You & Kim, 2020*; *Yu et al., 2005*). When no predesigned oligos are available, however, it is necessary to develop new ones. Oligo sensitivity is a trade-off between the specificity of the oligo to the DNA template and allowing some oligo-template mismatch if targeting different organisms in order to get an even coverage of all the representative organisms (*Parada, Needham & Fuhrman, 2016*). Depending on the user needs, there are many web-tools and software packages freely available for the oligo design. Some of the most widely used tools for primer design are Primer3 and its web interface Primer3Plus (*Untergasser et al., 2007*; *Untergasser et al., 2012*), Oligo7 (*Rychlik, 2007*) and Primeclade (*Gadberry et al., 2005*). To target unknown genes where only the protein or related gene sequences are known, it is necessary to design degenerate oligos. The latter take advantage of the codon degeneracy property of the amino acid sequences and, having degenerate bases in their sequences, represent a pool of unique primers that target the same amino acid coding sequence. Primer design tools for degenerate primers can require the input of proteins, such as CODEHOP (*Rose, Henikoff & Henikoff, 2003*; *Boyce, Chilana & Rose, 2009*) or Primer Premier (*Singh et al., 1998*); or the input of DNA sequences or alignments, such as DegePrimer (*Hugerth et al., 2014*), HYDEN (*Linhart & Shamir, 2005*) or FAS-DPD (*Iserte et al., 2013*).

Environmental communities pose many challenges for the oligo specificity as we often do not know what organisms are present and therefore it is difficult to foresee the possible nonspecific products (*Morales & Holben, 2009*; *Deiner et al., 2017*). *In silico* PCR is an essential step towards the design of specific oligos (*Yu & Zhang, 2011*). Some of the commonly used tools are UCSC In-Silico PCR (*Kent et al., 2002*), FastPCR (*Kalendar, Lee & Schulman, 2009*) and Primer-BLAST (*Untergasser et al., 2012*). The latter allows one to
check the oligo specificity against the comprehensive NCBI databases (*Sayers et al., 2020*). Further to their taxonomic specificity, oligos need to be tested for different parameters, such as the absence of homopolymer regions or di-nucleotide repetitions and the presence of a GC clamp (*Elbrecht, Hebert & Steinke, 2018*). Primers must also be scanned for the presence of secondary structures such as self-dimers, cross-dimers and hairpins (*Chuang, Cheng & Yang, 2013*). The analysis of secondary structure ΔG is integrated in the pipeline of widely used oligo design software, such as Oligo 7 (*Rychlik, 2007*) and Primer3 (*Untergasser et al., 2012*), or can be performed with specific software, such as PrimerROC (*Johnston et al., 2019*). The characteristics of the targeted organisms must also be taken in consideration. For instance, prokaryotic genomes rarely have introns as gene splicing is rare in these organisms (*Sorek & Cossart, 2010*), whereas introns and multiple splicing sites are widely present in eukaryotic genomes and must be taken in consideration when designing primers (*Goel, Singh & Aseri, 2013*; *Shafee & Lowe, 2017*).

In case the PCR target is a gene possessed only by a specific organism, the primers can be designed directly on that gene sequence. If more than one gene variant needs to be amplified (e.g., multiple species are targeted), a consensus sequence can be calculated and the oligos can then be designed on it (consensus primers). A consensus sequence is created from a sequence alignment and is defined as a sequence that reports the most frequent base present in the alignment in each position. The construction of this sequence, and consequently the designed oligos, is greatly influenced by the selection of the initial sequences. This pivotal step is usually not implemented in the oligo design software as these require the upload of preselected sequences. To date, only ARB implemented a toolkit that allows the creation of new primers and probes on sequences selected from the ARB phylogenetic tree of ribosomal sequences (*Ludwig et al., 2004*; *Essinger et al., 2015*). However, in order to work on other DNA portions, the user needs to create a sequence database to import inside the software.

Other tools, such as Morphocatcher (*Shirshikov, Pekov & Miroshnikov, 2019*) and Uniqprimer (*Karim et al., 2019*), propose high specificity primers. This is achieved by comparing the sequences that are the target of the PCR amplification with non-target sequences. However, no help in the sequence selection through phylogenetic tree visualization is available. Therefore in most tools, prior to the oligo design, the user has to retrieve the sequences of interest from a database (e.g., NCBI database), making sure that the sequences represent the DNA portion of interest and that they cover the same sequence fragment. This process can be complex and time-consuming especially when working with environmental microbial communities or working with a ubiquitous and divergent gene.

We present PhyloPrimer, a user-friendly and comprehensive online platform to (i) select the DNA sequences to use for oligo design, (ii) construct a consensus sequence, (iii) design microbial oligos (i.e., primers), (iv) test for oligo specificity through *in silico* tests and (v) test the oligos for the presence of secondary structures with the nearest-neighbour (NN) model for nucleic acids. In addition it provides a unique platform to check oligos (i.e., primer pairs, primer and probe assays, and single oligos) for both secondary structure and non-specific targets. The real strength of PhyloPrimer is the DNA sequence selection where the user can explore the diversity of the sequence of interest through a dynamic phylogenetic

tree. The sequences used for the tree construction are retrieved from a modified version of the GenBank database (*Sayers et al., 2019*) and are used by the software to increase taxon-specificity (down to the species level) of the designed oligos (positive/negative consensus approach). PhyloPrimer can design both primers and probes for PCR and qPCR applications, however, no qPCR-related tests were performed.

To test the efficiency and usability of PhyloPrimer we used the *rpoB* gene, which is a universal gene and encoding the *β*-subunit of RNA polymerase (*Adékambi, Drancourt & Raoult, 2009*). This is an essential enzyme to all the transcription processes in a cell as it accounts for the synthesis of mRNA, tRNA and rRNA. Its sequence is less conserved across different genomes compared to the 16S rRNA gene. This makes it less suitable to design universal primers but more suitable to design primers that can target specific organisms (*Case et al., 2007*). We tested PhyloPrimer by designing PCR primers suitable to the detection of organisms belonging to the *Streptococcus* genus and specific *Streptococcus* species (*Streptococcus agalactiae*, *Streptococcus pneumoniae*, *Streptococcus pyogenes*, *Streptococcus mutans* and *Streptococcus mitis*), amplifying taxon-specific *rpoB* genes from known mock communities.

## MATERIALS AND METHODS

### Implementation

PhyloPrimer runs on a remote server provided from the University of Bristol. The current server has 48 CPUs (64-bit Intel(R) Xeon(R) CPU E5-2680 v3 at 2.50 GHz). Only 4 PhyloPrimer processes at one time are allowed on the server, the excess processes enter a queue. On average, the oligo design requires 40–50 min whereas the oligo check requires 5–10 min. The web interface was implemented in HTML and JavaScript. PhyloPrimer is coded in Perl, JavaScript, HTML, CSS and MySQL. Two JavaScript packages were used: a modified version of PhyloCanvas v 1.7.3 (http://phylocanvas.org) and CanvasJS v 2.3.2 (https://canvasjs.com). The user can access PhyloPrimer through a web platform at https://www.cerealsdb.uk.net/cerealgenomics/phyloprimer. PhyloPrimer was tested and implemented using the Safari, Firefox, Chrome browsers. The website uses General Data Protection Regulation (GDPR) cookie acceptance box on the first use. All the PhyloPrimer scripts are also available through the PhyloPrimer GitHub page (https://github.com/gvMicroarctic/PhyloPrimer).

### General workflow and dynamic selection

The PhyloPrimer web platform is structured with sequential web pages that can be categorized into four different groups: (i) the home page, (ii) the input pages, (iii) the oligo pages and (iv) the result page. From the home page, the user can select one of the three different input pages available for uploading the data (e.g., DNA sequences, DNA alignments and Newick trees) where each page corresponds to a different modality to use PhyloPrimer. Once the data are uploaded, the user is redirected to the oligo pages where there are different parameter settings to design either primer assays, primer and probe assays or single oligos. Once the user submits these parameters, the oligo design and the oligo check are performed on the web server. As soon as PhyloPrimer has finished the
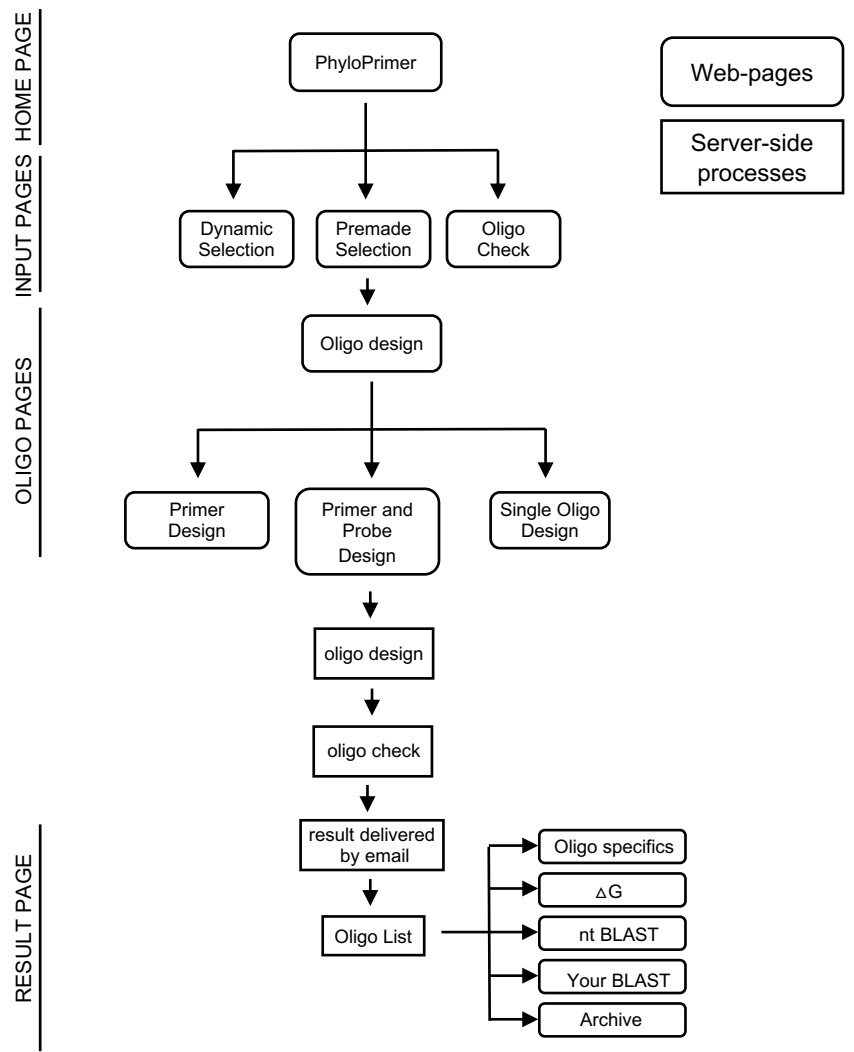

**Figure 1** **PhyloPrimer structure indicating the web pages and the server-side processes.**

analyses, the user receives an email with a link to the result page where the user can explore the designed oligos and choose the ones which will be used for future work (Fig. 1).

PhyloPrimer can be used in three different modalities. It can be used to design oligos from DNA sequences interactively selected from a dynamic phylogenetic tree (Dynamic Selection; Fig. 2A), to design oligos from preselected DNA sequences (Premade Selection; Fig. 2B) and to check predesigned oligos (Oligo Check; Fig. 2C). The Dynamic Selection modality is the strength of PhyloPrimer and was developed to facilitate the selection and retrieval of NCBI sequences for the oligo design. The processes reported in the rest of the manuscript describe this modality and details on the others can be found in the manual at https://github.com/gvMicroarctic/PhyloPrimer.

The user can upload one or more sequences representing the same DNA portion (e.g., same gene or gene fragment belonging to different organisms). PhyloPrimer then runs

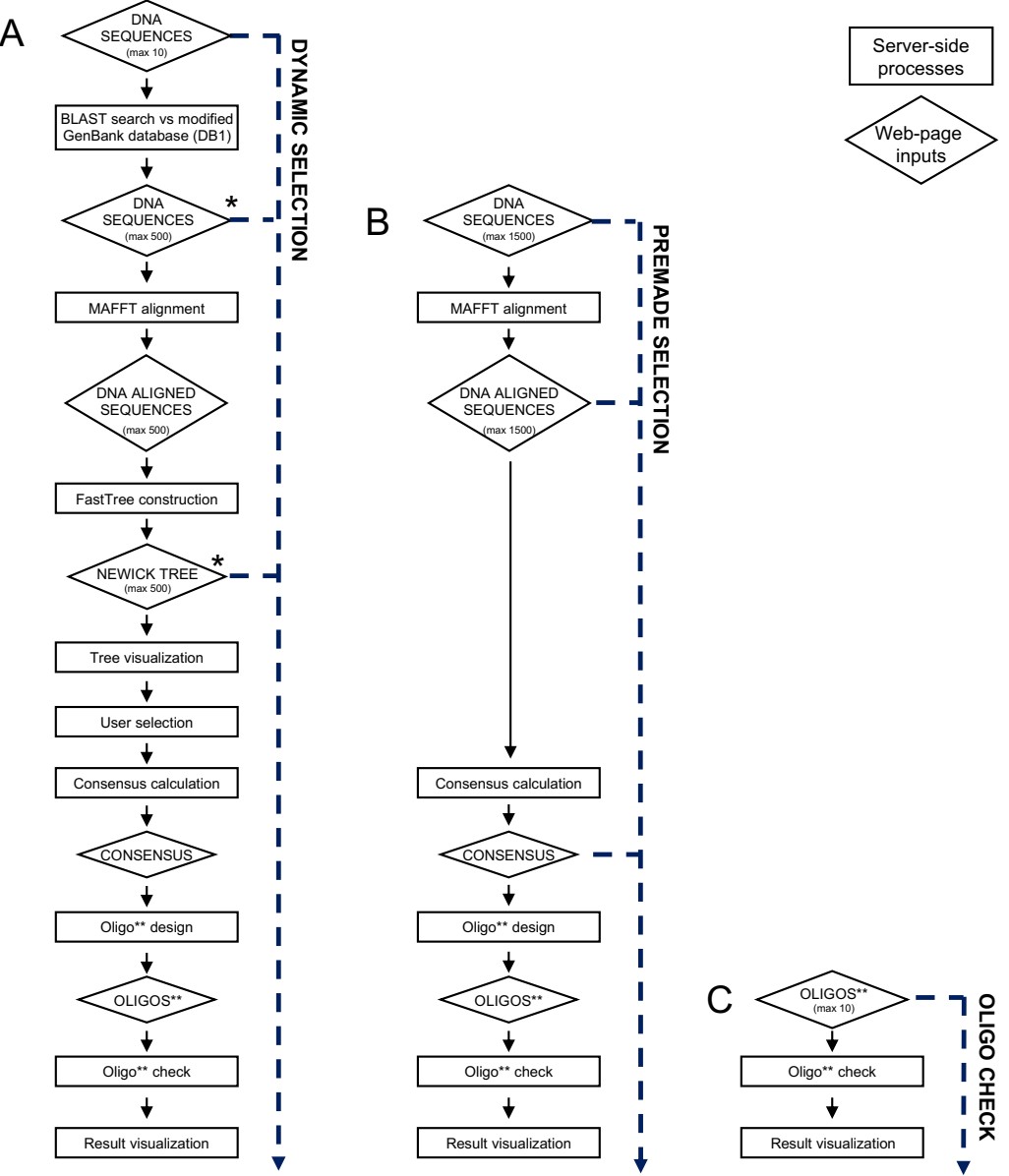

**Figure 2** **Detailed scheme of the three different input pages and workflows.** The three workflows are Dynamic Selection (A), Premade Selection (B) and Oligo Check (C). Through the Dynamics Selection page the user can input three different kind of data: up to 10 genes or DNA regions of interest, up to 500 DNA sequences and a Newick tree (together with an alignment file). The Premade Selection page permits the uploading of up to 1,500 DNA sequences, 1,500 DNA aligned sequences or directly the consensus sequence that will be used for the oligo design. In the Oligo Check page only the upload of predesigned oligos is allowed. Different processes on the server-side of PhyloPrimer will start in relation to which data was uploaded. *There can be optional input files for taxonomy and protein information. If a Newick file is the input, an additional alignment file must be uploaded. ** Oligos are intended as primers pairs, primer pairs plus a probe or single oligos.

MegaBLAST (*Morgulis et al., 2008*; *Baxevanis, 2020*) against the database DB1 (details in section "Databases"). The user can set up three BLAST parameters: the e-value (the probability of finding a match by chance), the identity percentage (the percentage of bases shared between the query and the subject sequence), and coverage percentage (the percentage of bases of the query sequence that are covered by the subject sequence). If more than four matching sequences were found in the database, PhyloPrimer runs a MAFFT alignment (*Katoh & Standley, 2013*) and then constructs a phylogenetic tree with FastTree (*Price, Dehal & Arkin, 2009*). The user can explore the dynamic tree and look at the sequence information connected to each retrieved GenBank entry (e.g., taxonomy). The user can then select, in the tree, the sequences that must be used for the consensus calculation and therefore the oligo calculation (Fig. 3A).

## Consensus calculation and primer specificity

PhyloPrimer uses a consensus approach for the oligo design or, in other words, it designs the oligos from a consensus sequence. After the user selects the adequate sequences from the dynamic tree, PhyloPrimer calculates two consensus sequences. The positive consensus is the consensus calculated from the selected sequences, and the one used for the oligo design. The negative consensus is calculated from the sequences that were not selected from the tree and is used to increase the oligo specificity to the target organisms looking at the base difference between the two consensus sequences (Fig. 3B). After the consensus construction, Phyloprimer compares the two sequences and finds the differing positions. To create taxon-specific oligos, PhyloPrimer uses this information when scoring the oligos with the aim to retrieve the best ones to be visualized in the dynamic result page (Fig. 3C). To guarantee a high level of oligo-specificity, PhyloPrimer also runs an *in silico* BLAST search where oligos that are specific for the targeted organisms are selected. The user can specify if the oligos must be species-, genus-, family-, order-, class-, phylum- or domain-specific. PhyloPrimer picks which are the organisms of interest from the phylogenetic tree selections.

## Oligo design and scoring system

The consensus sequence can be uploaded to PhyloPrimer by the user through the Premade Selection page or it can be calculated by PhyloPrimer itself. The software constructs the consensus with the DNA sequences or alignments uploaded through the Premade Selection page or with the sequences that were selected by the user on the dynamic tree (Dynamic Selection mode). In order for PhyloPrimer to find suitable oligos, the consensus must have one or more conserved regions, DNA regions that are in common among all the selected/uploaded sequences. If no conserved regions are present, the consensus sequence will be represented by long stretches of degenerate bases and the software will not be able to design any oligo from it. There can be different reasons for this: (i) the sequence selection was too broad for the target gene family, (ii) the selected sequences did not include only sequences from the same gene family, (iii) the sequences represented different DNA regions of the same gene or (iv) the studied gene family is very divergent. In general, it is more likely to have a conserved region in the consensus when working with closely related sequences, for example, when developing oligos for a specific species rather than for an

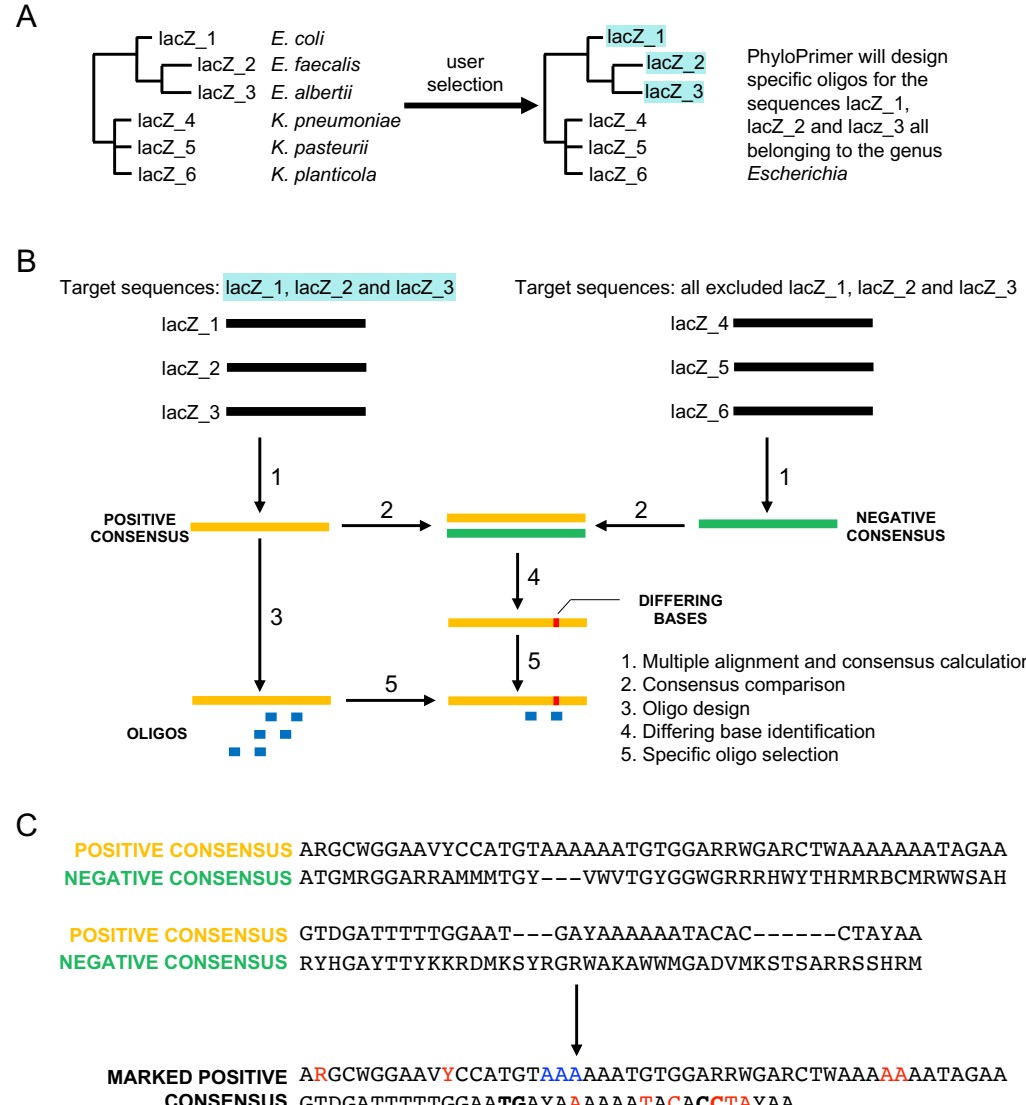

**Figure 3 PhyloPrimer positive and negative consensus workflow.** In the Dynamic Selection mode, the consensus design starts with the selection of DNA sequences on the phylogenetic tree (A). Successively, the selected sequences (i.e., lacZ_1, lacZ_2 and lacZ_3) are used to construct the positive consensus and the others are used to calculate the negative consensus (i.e., lacZ_4, lacZ_5 and lacZ_6). The two consensus sequences are compared, positive consensus areas with differing bases are identified and oligos specific to that area are selected (B). In any consensus visualization of the consensus sequence, PhyloPrimer reports only the positive consensus with marked letters where differing with the negative sequence (C). The base color code is as follows: red letters indicate positions where the two sequences presented differing bases, blue letters indicate positions where there are bases on the positive consensus but gaps in the negative and bold letters flank regions where there were gaps on the positive consensus but bases on the negative consensus. A degenerate base is marked as differing only if that base does not contain the corresponding base of the negative consensus.

entire gene family. However, when the aim is to develop oligos at gene level, the presence of a conserved gene region between different organisms highly depends on the gene sequence. It is essential to know the gene family object of the study and to check the consensus sequence that PhyloPrimer reports. In case the consensus presents a lot of degeneracy, it will be necessary to adjust the maximum number of degenerate bases allowed inside the oligo sequence in the oligo design pages. If this does not help, the design of different oligos for different cluster of organisms should be considered.

In PhyloPrimer the conserved region of the consensus sequence is determined by the maximum number of degenerate bases that is allowed inside the oligo sequences. For instance, if the user sets the maximum degenerate base value to 1, PhyloPrimer will discard all the oligos that have more than 1 degenerate base in the sequence or, in other words, won't consider the areas of the consensus that have an incidence of degenerate bases higher than 1 base every oligo length (between 18–22 bp by default).

PhyloPrimer will start the oligo design only once the positive consensus has been obtained. For each possible oligo length, the software extracts from the consensus sequence all the possible subsequences of that length (Fig. 4A). This first step creates the starting pool of oligos that the following steps will check and discard if not respecting all the design parameters. The first check step discards by default the oligos that are not unique in the consensus sequence, that have homopolymer repetition longer than 3 bases, dinucleotide repetition longer than 6 bases, a GC content lower than 40% or higher than 60%, and will check and discard the oligos that do not have between 2 and 4 Gs/Cs in the last 5 bases of 3′ oligo end (GC clamp). PhyloPrimer will also check if the oligos have a higher number of degenerate bases than the limit and that only the correct degenerate bases are present (all except from N by default). The default number of degenerate bases is set by PhyloPrimer in relation to how many degenerate bases were found inside the consensus sequence but can be changed by the user (Fig. 4B).

PhyloPrimer then calculates the reverse complement of all the oligos and considers the original oligos as putative forward primers and the oligo reverse complements as putative reverse primers (Fig. 4C). All the forward and reverse primers are progressively checked to have a valid melting temperature (between 54 °C and 64 °C by default) and, in case the presence of degenerate bases is allowed, not to have degenerate bases in the last 5 bases of the 5′ oligo end and last 2 bases of the 3′ end oligo tails (by default). The software also checks for the presence of self-dimer and hairpin secondary structures and discards any oligos with a secondary structure associated to a $\Delta G$ value lower than -5 and -3 kcal mol$^{-1}$, respectively (Fig. 4D). Primers meeting the above criteria are then considered as suitable primer pairs (Fig. 4E). The primer pairs are first selected considering the distance between their 5′ ends on the consensus (between 200 and 600 bases by default). The primer pairs are then discarded if the melting temperature difference between forward and reverse primers is higher than 5 °C or the annealing temperature does not range between 50 °C and 60 °C (Fig. 4F).

At this point, all the remaining primer pairs have all the requirements that were set by the user through the oligo pages. All the following steps aim to retrieve the best primer pairs that will be visualized in the result page. This is achieved by assigning points to each

Processes | oligo design | oligo check | oligo scoring

| | |
|---|---|
| **A** | For each possible oligo length, the consensus is chopped into n oligos with a +1 shift |
| **B** | All the oligos are checked for:<br>• Homopolymers<br>• Dinucleotide repetitions<br>• Degeneracy<br>• GC%<br>• Uniqueness in the consensus |
| **C** | All the oligos are converted into their reverse complement<br><br>Forward primer = oligo        Reverse primer = reverse complement of the oligo |
| **D** | All the forward primers are checked for:<br>• $T_m$<br>• Degenerate bases in tails<br>• GC clamp<br>• Self-dimer $\triangle G$<br>• Hairpin $\triangle G$<br><br>All the reverse primer are checked for:<br>• $T_m$<br>• Degenerate bases in tails<br>• GC clamp<br>• Self-dimer $\triangle G$<br>• Hairpin $\triangle G$ |
| **E** | Create primer pairs |
| **F** | All the primer pairs are checked for:<br>• $T_a$<br>• $T_m$ difference between forward and reverse primers |
| **G** | PhyloPrimer assign points to each primer pair following this criteria:<br>• Differing bases in the second to last base at the 3' end: +10<br>• Differing bases in the last base at the 3' end: +20<br>• Differing bases in the rest of the oligo sequence: +2<br>• $T_m$ difference between forward and reverse primers is lower than 1 °C: +1<br>• The $\triangle G$ is higher than -1 kcal mol$^{-1}$: +1 (for hairpins and self-dimers)<br>• Degenerate bases: -2 if R, Y, S, W, K and M, -3 if B, D, H and V and -4 if N |
| **H** | The first best 1000* primer pairs are checked for cross-dimer formation |
| **I** | The first best 500* primer pairs are checked with a BLAST and Bowtie search against the database DB2 |
| **J** | The BLAST results are retrieved and the points are assigned to each primer pair as follows:<br>• Species that was selected in the dynamic tree: +10<br>• Species** that was selected in the dynamic tree and has been found for the first time: +20<br>• Species** which was not selected in dynamic tree: -40 |
| **L** | The first 100 primer pairs are retrieved and visualized in the result page |

**Figure 4** **Primer design workflow.** Oligo design, check and scoring processes are indicated. One asterisk (*): 250 if no negative consensus was present, no differing bases between the two consensus sequences were present or no differences were taken in consideration in the scoring system. Two asterisks (**): depending on the visualization criteria that were selected, +20 and −40 points are assigned if the different oligos BLAST searched against DB2 entries belonged to genera, families, orders, classes, phyla and domains that were selected from the phylogenetic tree.

primer pair as follows: 1 point is assigned to the primer pair if the melting temperature of the forward and reverse primers differ by less than 1 °C, for each secondary structure 1 point is assigned if the $\Delta G$ value is higher than $-1$ kcal mol$^{-1}$. Moreover, 20 points are assigned if a base polymorphic between the positive and the negative consensus is present in the last$\Delta G$ base of the 3′ end and 10 points if it is present in the penultimate base. Two points are also assigned for each additional base difference between the positive and the negative consensus (Fig. 4G). The best 1,000 primer pairs are selected, checked for cross-dimer formation and discarded if the $\Delta G$ values are lower than $-5$ kcal mol$^{-1}$ (Fig. 4H).

PhyloPrimer selects the first 500 primer pairs that scored the highest points according to the scoring system (Fig. 4I). The oligos belonging to those first 500 primer pairs are BLAST searched against DB2 (details in section "Databases"). PhyloPrimer then screens the BLAST results and retrieves the database sequences that matched both to the forward and reverse primers and uses them to perform a global alignment with Bowtie (*Langmead, 2010*). PhyloPrimer then checks the alignment results and considers only the database sequences that were matched by both the forward and the reverse primers of a primer pairs. If that sequence belongs to one of the species that were selected from the dynamic tree, PhyloPrimer assigns 10 points to the primer pair, if the species was not among the selected species it deducts 40 points, and every time there is a new correct species PhyloPrimer adds 20 points to the total. By default, PhyloPrimer does not assign more points to primers that belong to the same genus (or higher ranks) of the selected tree entries. But if these visualization parameters are checked, PhyloPrimer will assign 20 points to the entries that belong to the same taxonomy and deduct 40 to those that do not. This is for facilitating the design of oligos that are specific to a genus (or higher taxonomic group) rather than only specific to certain species. In case an additional file was uploaded by the user for an additional BLAST check, PhyloPrimer will also BLAST all the oligos against that database but the outcome will not be the object of the scoring system (Fig. 4J).

The described scoring criteria are all active by default but any of those can be deselected by the user on the Oligo Design page. PhyloPrimer then selects the first 100 primer pairs and these primer pairs will be the ones showed in the last Result Page. When degenerate bases are present inside the oligo sequences, the melting temperature and the GC content are calculated as the mean of these values in each of the possible oligo (Fig. 4L).

The design process for primer pair/probe assays and single oligos is very similar to that described above and is described fully in the software manual.

## Melting temperature and $\Delta G$ secondary structures

PhyloPrimer calculates oligo melting temperatures ($T_m$) and secondary structure Gibbs free energies ($\Delta G$) with the nearest-neighbor (NN) model for nucleic acids. This model predicts the thermodynamic behavior of a DNA molecule using the thermodynamic parameters of each nucleotide pair composing the molecule itself. Both the $T_m$ and the $\Delta G$ calculation rely on the use of the thermodynamics parameters enthalpy ($\Delta H$) and entropy ($\Delta S$). These parameters were derived from calorimetry and spectroscopic experiments of DNA duplexes for nucleotide base pair motives (*SantaLucia & Hicks, 2004*), internal
_______________________________________________

mismatches (*Allawi & Santalucia, 1997*; *Allawi & SantaLucia, 1998a*; *Allawi & SantaLucia, 1998b*; *Allawi & SantaLucia, 1998c*; *Peyret et al., 1999*), dangling ends (*Bommarito, Peyret & SantaLucia, 2000*) and hairpin terminal mismatches (unpublished data). The latter were retrieved from the UNAFold database (*Markham & Zuker, 2008*). The $\Delta H$ and $\Delta S$ are considered temperature independent when working with nucleic acids and are reported for 1 M $Na^+$ conditions.

The melting temperature ($T_m$) of a DNA molecule is the temperature in which half of the DNA is paired with its complement and half is single-stranded. The correct calculation of this parameter is essential to the correct calculation of the PCR annealing temperature, and it is pivotal for the qPCR probe when wanting to differentiate amplicon expression levels. PhyloPrimer calculates $T_m$ with the formula reported in *SantaLucia & Hicks (2004)*. The annealing temperature, $T_a$, is calculated as the lowest melting temperature (if more than one oligo is present) minus 5. This is an indicative calculation as the optimal annealing temperature can considerably vary in relation to the polymerase that is used during the PCR.

The $\Delta G$, or Gibbs free energy, estimates if a reaction can occur spontaneously ($\Delta G$ lower than 0, exergonic reaction) or not ($\Delta G$ higher than 0, endergonic reaction) and therefore indicates how stable a particular DNA structure is at a certain temperature. In this case, $\Delta G$ represents the quantity of energy needed to fully break a secondary structure. The lower it is (more negative), the more stable and likely to occur the secondary structure will be and the more energy will be required to break it. $\Delta G$ is defined as equal to the enthalpy minus the product of the temperature times the entropy (Gibbs free energy equation). PhyloPrimer calculates the $\Delta G$ for three different secondary structure formations: self-dimers (i.e., dimers formed within the oligo itself), cross-dimers (i.e., dimers formed between different oligos) and hairpin loops (i.e., hairpin-like secondary structures formed within the oligo itself). For each of these different structures, different rules must be applied to $\Delta H$ and $\Delta S$ calculation which are then used to apply the Gibbs free energy equation (*SantaLucia & Hicks, 2004*).

Melting temperature and $\Delta G$ values obtained in this way (*SantaLucia & Hicks, 2004*; Gibbs free energy equation) are valid only in 1 M $Na^+$ condition. Because the PCR conditions can span a wide range of different conditions, salt correction formulas must be applied to correct the obtained values (*Owczarzy et al., 2004*; *Owczarzy et al., 2008*). Depending on the polymerase used and the PCR protocol, $Mg^{2+}$ and monovalent ions can vary considerably and rarely the 1 M $Na^+$ condition is respected. Therefore, PhyloPrimer performs salt-correction correction with the parameters reported and customized in the oligo pages therefore calibrating the corrections on the user specific PCR conditions.

When dealing with degenerate oligos, PhyloPrimer calculates melting temperature and $\Delta G$ values for all the possible oligos. The final $T_m$ is the average of all the calculated $T_m$ whereas the final $\Delta G$ is the lowest $\Delta G$. More information on the $T_m$ calculation, $\Delta G$ calculation and correction formulas, together with all the thermodynamic parameters, can be found in the manual.

## Databases

PhyloPrimer uses external nucleotide sequence databases in two points of the pipeline. The first point is when it BLAST searches the sequences uploaded in the Dynamic Selection mode to retrieve similar sequences and construct a dynamic phylogenetic tree (DB1), and the second when it checks the oligo specificity through *in silico* PCR (DB2). DB1 is constituted by protein, rRNA, tRNA and tmRNA coding regions annotated from GenBank prokaryotic genomes (*Sayers et al., 2019*). Nucleotide sequences from a maximum of 50 different genome assemblies or complete genomes are reported per organism for a total of 78,710 bacterial genomes and 3,247 archaeal genomes. DB2 is the nucleotide database (ftp://ftp.ncbi.nih.gov/blast/db/FASTA/nt.gz) which contains partially non-redundant nucleotide sequences from the GenBank, EMBL and DDBJ databases. The sequence taxonomy of DB1 and DB2 relies on GenBank genome taxonomy (*Benson et al., 2018*). The two databases can be downloaded from the PhyloPrimer GitHUB page (https://github.com/gvMicroarctic/PhyloPrimer). DB1 and DB2 were last updated in April 2021 and are updated every two months. At the moment of the publication they contained 289,757,008 (DB1) and 68,965,867 (DB2) entries. The databases cannot be substituted. However, the user can upload extra sequences for the *in silico* check; in this case PhyloPrimer will check the taxon-specificity of the oligos against both the DB2 sequences and the user uploaded sequences.

## PhyloPrimer test

The primer pairs were designed to amplify all the organisms related to the genus *Streptococcus* (PP1), and five *Streptococcus* species: *Streptococcus agalactiae* (PP2), *Streptococcus pneumoniae* (PP3), *Streptococcus pyogenes* (PP4), *Streptococcus mutans* (PP5) and *Streptococcus mitis* (PP6) (Table 1). The primer design was performed with PhyloPrimer (Dynamic Selection mode) where six *rpoB* gene sequences were uploaded for the tree construction (Data S1): one for each *Streptococcus* species in the mock communities and one sequence belonging to *S. dysgalactiae* which was shown to be highly related to *S. pyogenes* (*Jensen & Kilian, 2012*). The primers were designed with default parameters with exception of melting and annealing temperature (60−75 °C), monovalent ion concentration (0 mM), magnesium ion concentration (2.5 mM), oligo concentration (0.6 μM) and dNTP concentration (1.2 μM) which were modified accordingly to the specifics of the polymerase used for the PCR. Furthermore, in order to be sure the DNA was amplifiable in all the mock communities, the primers 341F and 518R were also used to amplify the 16S rRNA gene as a positive control (Table 1) (*Muyzer, De Waal & Uitterlinden, 1993*).

The primers were tested with four mock communities: Metagenomic Control Material for Pathogen Detection (ATCC® MSA-4000), 10 Strain Staggered Mix Genomic Material (ATCC® MSA-1001), Skin Microbiome Genomic Mix (ATCC® MSA-1005) and ZymoBIOMICS Microbial Community DNA Standard (D6306, Zymo Research). These communities comprise several organisms, present with different abundances and ranging in microbial diversity. In the following tests they will be called community A, B, C and D, respectively (Table 2).

**Table 1** **Primer specifics.** All the primers were designed with PhyloPrimer web platform except the 16S rRNA primers which were designed by *Muyzer, De Waal & Uitterlinden (1993)*.

| Primers | Primer sequences | |
|---|---|---|
| | Forward | Reverse |
| 16S rRNA | CCTACGGGAGGCAGCAG | GGCACAGCCTGACGTTGCAT |
| PP1 | TTGACWCGTGACCGTGCTGG | GGCACAGCCTGACGTTGCAT |
| PP2 | GCGTCGCGAAGATGGTTCT | ACCTCAGCACCAATGCGGATGA |
| PP3 | AGCTTGCTTGTRGCTCGCTT | CTCAGTCACAACGGCTGCATCG |
| PP4 | CAGTTGCACAGGCCAATTCGA | GTGAGCCATCTTGACGACGGAT |
| PP5 | GCGAGCGTCTTGTCAAGGAT | ACCACCAAGCGGCTGTTGA |
| PP6 | ACATGCAACGTCAGGCTGT | AGTACGAGCAGCCATACCAAGG |

| Primers | Target organisms | Amplicon size (bp)[*] |
|---|---|---|
| 16S rRNA | Bacteria | 200 |
| PP1 | *Streptococcus spp.* | 470 |
| PP2 | *Streptococcus agalactiae* | 410 |
| PP3 | *Streptococcus pneumoniae* | 270 |
| PP4 | *Streptococcus pyogenes* | 380 |
| PP5 | *Streptococcus mutans* | 870 |
| PP6 | *Streptococcus mitis* | 1000 |

**Notes.**
[*]PhyloPrimer predicted amplicon length

Each mock community DNA was used as template for the PCR amplification using the primers 16S rRNA and the PhyloPrimer developed primer pairs (PP1, PP2, PP3, PP4, PP5 and PP6). The 25 µL PCR solution consisted in 12.5 µL for 2X KAPA HiFi HotStart ReadyMix polymerase (KAPA BIOSYSTEMS), 1.5 µL of 5 µM forward primer, 1.5 µL of 5 µM reverse primer, between 1–3 µL of template DNA (corresponding to 4 ng of DNA) and nuclease-free water up to volume. A negative control where the template DNA was substituted with nuclease-free water was included for every primer pair.

The PCR was performed using an Eppendorf Mastercycler nexus X2 thermal cycler (Eppendorf) with the following conditions: 95 °C for 3 min, 25 cycles of 98 °C for 20 s, 64 °C for 15 s and 72 °C for 20 s, and a final extension of 72 °C for 1 min. The annealing temperature of 64 °C was used for all the primer pairs PP1, PP2, PP4, PP5 and PP6, whereas we used 65 °C for PP3 and 62 °C for the 16S rRNA primers.

For each sample, 6 µL of PCR product was then run with 2 µL of gel loading buffer (NEB) on 1.5% w/v horizontal agarose gel (0.5 mg) ethidium bromide ml$^{-1}$ in 1x TEA buffer (Tris acetate EDTA) and run for 30 min at 120 mV (Bio-Rad PowerPac 300, Bio-Rad Laboratories). Gel pictures were visualized under UV light and captured with GelDoc-ItTS2 Imager (UVP). No bands were shown in any of the negative control lanes. GelPilot 100 bp Plus Ladder (Qiagen) was run for amplicon size comparison.

The non-specific amplicon band obtained in community A with the primer pair PP6 was sequenced with the nanopore technology. The library preparation was performed using the SQK-LSK109 kit (Oxford Nanopore Technologies, Oxford, UK). The sequencing was

Table 2 **Mock microbial composition.** Composition for the communities A, B, C and D where community A corresponds to ATCC® MSA-4000, B to ATCC® MSA-1001, C to ATCC® MSA-1005 and D to the ZymoBIOMICS community.

| Species | Relative abundance (%) | | | |
|---|---|---|---|---|
| | A | B | C | D |
| *Acinetobacter baumannii* | 0.10 | – | – | – |
| *Acinetobacter johnsonii* | – | – | 16.70 | – |
| *Bacillus cereus* | – | 4.48 | – | – |
| *Bacillus subtilis* | – | – | – | 12.00 |
| *Bifidobacterium adolescentis* | – | 0.04 | – | – |
| *Clostridium beijerinckii* | – | 0.45 | – | – |
| *Corynebacterium striatum* | – | – | 16.70 | – |
| *Cryptococcus neoformans* | – | – | – | 2.00 |
| *Cutibacterium acnes* | – | – | 16.70 | – |
| *Deinococcus radiodurans* | – | 0.04 | – | – |
| *Enterococcus faecalis* | 0.70 | 0.04 | – | 12.00 |
| *Escherichia coli* | 1.40 | 4.48 | – | 12.00 |
| *Klebsiella pneumoniae* | 14.40 | – | – | – |
| *Lactobacillus fermentum* | – | – | – | 12.00 |
| *Lactobacillus gasseri* | – | 0.45 | – | – |
| *Listeria monocytogenes* | – | – | – | 12.00 |
| *Micrococcus luteus* | – | – | 16.70 | – |
| *Neisseria meningitidis* | 28.90 | – | – | – |
| *Pseudomonas aeruginosa* | 0.30 | – | – | 12.00 |
| *Rhodobacter sphaeroides* | – | 44.78 | – | – |
| *Saccharomyces cerevisiae* | – | – | – | 2.00 |
| *Salmonella enterica* | – | – | – | 12.00 |
| *Staphylococcus aureus* | 15.10 | – | – | 12.00 |
| *Staphylococcus epidermidis* | – | 44.78 | 16.70 | – |
| *Streptococcus agalactiae* | 2.90 | – | – | – |
| *Streptococcus mitis* | – | – | 16.70 | – |
| *Streptococcus mutans* | – | 0.45 | – | – |
| *Streptococcus pneumoniae* | 28.90 | – | – | – |
| *Streptococcus pyogenes* | 7.20 | – | – | – |

performed using a flow cell FLO-MIN106 with a MinION device and can be found in the European Nucleotide Archive (ENA) at EMBL-EBI under accession number PRJEB42474. The sequences were basecalled using Guppy v 3.2.2 (Oxford Nanopore Technologies). Sequences were then taxonomy assigned by BLAST search against genomes contained in community A. The list of the genomes used to create the BLAST database can be found in Table S1.

## RESULTS AND DISCUSSION

The development of taxonomic specific primers is essential to many environmental and biomedical biomonitoring and detection studies (e.g., *Ai et al., 2019*; *Dos Santos et al., 2020*;

*Liu et al., 2003*; *Song et al., 2000*) where the recent COVID-19 pandemic is a perfect example of how important is the design of species-specific primers to detect a specific organism of interest (*Park et al., 2020*). We developed PhyloPrimer, an automated platform that integrates a new pipeline which aims to design taxonomic-specific oligos and tests them for secondary structures and target specificity.

The 16S rRNA gene was amplified in all the four communities showing that all the DNA communities had amplifiable microbial DNA (Fig. 5). The primer pair PP1 which was specific for the *Streptococcus* genus produced amplicons of the expected size (about 500 bp) in all the communities except from community D where no *Streptococcus* species were present (Table 2). Primers PP2, PP3 and PP4 which targeted respectively *S. agalactiae*, *S. pneumoniae* and *S. pyogenes* showed PCR products only in community A which was the only community that contained these organisms. The amplicon size also reflected that predicted by PhyloPrimer being around 410, 270 and 380 bp for primer PP2, PP3 and PP4, respectively. The primer pair PP5 only amplified community B which was the only one containing *S. mutans*. Finally, the primer pair PP6, specific for *S. mitis*, showed bands (around 1,000 bp) in both community A and C. Community A did not contain *S. mitis* and therefore species specificity was not achieved with this primer pair. This non-specific band is due to the amplification of the other *Streptococcus* species present in community A (Fig. 5 and Table 2). Of the 268,551 amplified sequences that matched to genomes present in community A, in fact, more than 99% of the sequences was assigned to the genus *Streptococcus*. Of these, 66% of the sequence was assigned to *S. pneumoniae*, 33% to *S. pyogenes*, and 1% to *S. agalactiae*.

Organisms belonging to the species *S. mitis* have been previously observed to not form a well-isolated phylogenetic cluster (*Whatmore et al., 2000*). In particular, *S. mitis* has been shown to be closely related to *S. pneumoniae* (*Kawamura et al., 1995*). The close similarity between these two species can be observed from PhyloPrimer tree where *S. mitis* and *S. pneumoniae* organisms do not show well-separated phylogenetic clusters (https://www.cerealsdb.uk.net/cerealgenomics/cgi-bin/tree_paper.cgi). Therefore the non-specificity of the primer pair PP6 is caused by the scarce differentiation of *S. mitis* from the other *Streptococcus* species where, in particular, the non-specific band observed in community A could be due to the amplification of *S. pneumoniae* (Fig. 5). This result is confirmed by the PhyloPrimer *in silico* taxonomic test where all the designed primer pairs targeting *S. mitis* also targeted *S. pneumoniae*. Furthermore, the positive consensus sequence calculated for *S. mitis* only had a total of 8 differing bases with the negative consensus (Fig. S1); and no differing bases at all between positive and negative consensus within either its forward or reverse primer sequences (i.e., PP6) whereases all the other selected primer pairs (i.e., PP1, PP2, PP3, PP4 and PP5) had differing bases (Fig. S2). It was therefore not possible to design a species specific *rpoB* primer pair for *S. mitis* as primers could not be made to target any of the eight known bases unique to this species, due to design constraints.

PhyloPrimer performed well with different organism and gene settings and showed overall good results when tested on the mock communities. There are several oligo design software, such as MPrimer (*Shen et al., 2010*), PrimerDesign-M (*Yoon & Leitner, 2015*),

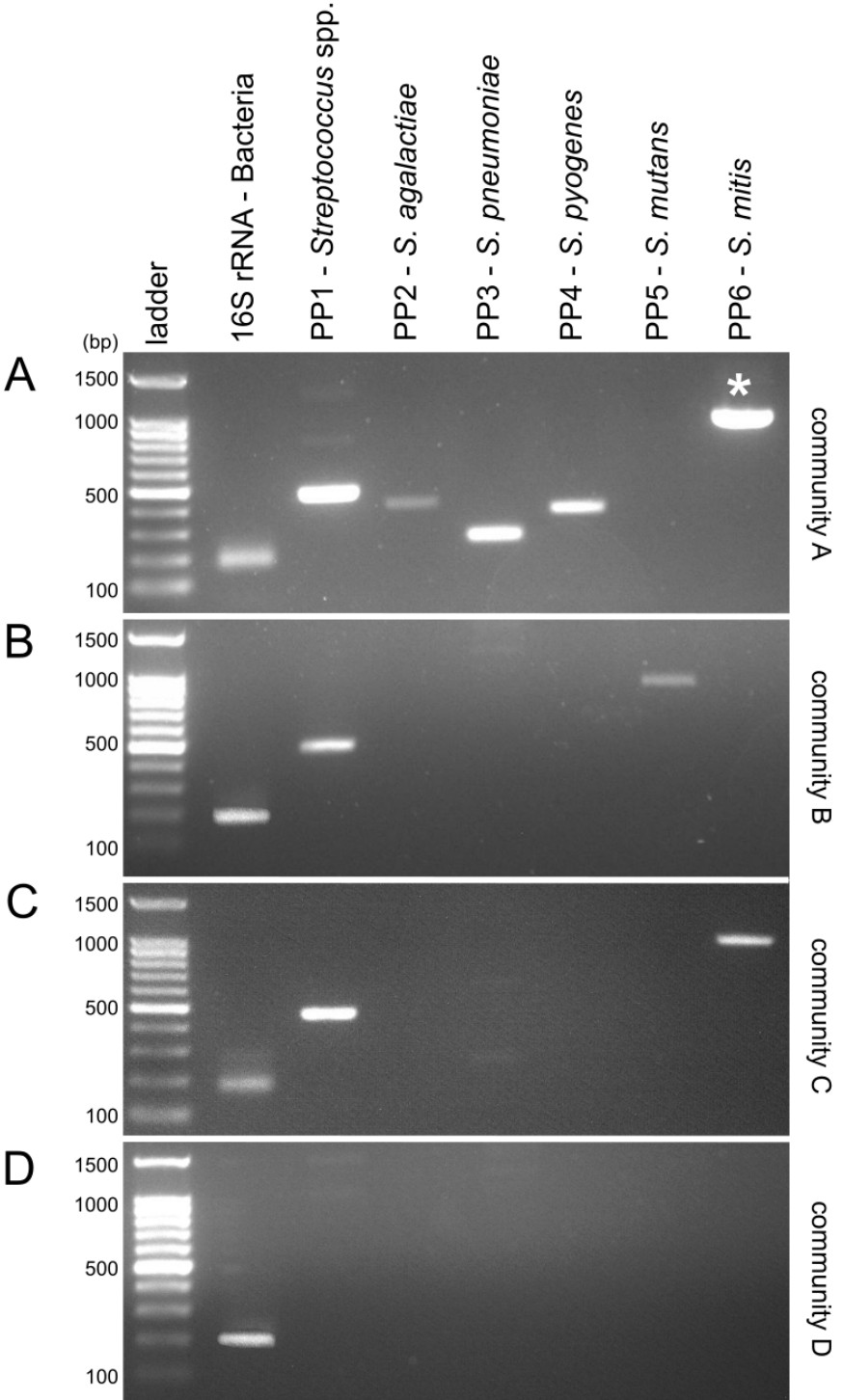

**Figure 5 Agarose gel pictures of the PCR products amplified with the 16S rRNA and the PhyloPrimer designed primer (PP1, PP2, PP3, PP4, PP5 and PP6) on the mock communities A, B, C and D.** The white star marks the non-specific band found in community A for the primer PP6. All the other primer pairs amplified only the expected communities and no false negatives occurred.

MRPrimerW (*Kim et al., 2016*) and Oli2go (*Hendling et al., 2018*), that are similar to PhyloPrimer at different stages of the pipeline: homolog screening, secondary structure check and oligo scoring. While others implement the use of a positive and negative consensus sequence for oligo design, such as in the case of Morphocatcher (*Shirshikov, Pekov & Miroshnikov, 2019*) and Uniqprimer (*Karim et al., 2019*), PhyloPrimer automizes all the steps, from homologous sequence selection to oligo scoring, providing a user-friendly oligo design platform.

The software also comes with some limitations. For example, the database used for the tree construction contains only microbial sequences, which lack the added complexity of lengthy intron-containing eukaryotic genes. DB1 is also constituted by coding-region sequences and therefore PhyloPrimer cannot build a phylogenetic tree with intergenic regions. Also, PhyloPrimer does not design degenerate oligos specifically. PhyloPrimer uses a consensus approach and it designs the oligos from a consensus sequence calculated from a DNA alignment. Therefore it will not introduce degeneracy on purpose and will design oligos containing degenerate bases only if present in the consensus sequence and if necessary to the design of suitable oligos.

## CONCLUSION

We developed PhyloPrimer, a semi-automated and user-friendly pipeline to go from sequence selection to oligo design, and in silico tested oligos. This tool aims to help with oligo design of complex environmental communities speeding up and providing a solid and reproducible pipeline for the oligo design and *in silico* tests. We demonstrated the relevance of this approach which showed good results in terms of oligo-specificity when tested on microbial mock communities.

## ACKNOWLEDGEMENTS

We thank Paul Wilkinson for the help given to set up the web server. We thank Ella Boswell and Nina Blampied, from the University of Bristol, and Layla Rahiem and Mirjam Pikaart, from Avans University, who used PhyloPrimer and gave useful feedback in return.

### Funding

This work was supported by the European Union's Horizon 2020 research and innovation programme under the Marie Skłodowska-Curie grant agreement No 675546 (MicroArctic network). Funding support was also provided by the Natural Environment Research Council grant NE/J02399X/1 awarded to GB. The funders had no role in study design, data collection and analysis, decision to publish, or preparation of the manuscript.

### Grant Disclosures

The following grant information was disclosed by the authors:

European Union's Horizon 2020 research and innovation programme under the Marie Skłodowska-Curie: 675546.
Natural Environment Research Council grant: NE/J02399X/1.

### Competing Interests

Cédric Malandain is employed by HYDREKA.

### Author Contributions

- Gilda Varliero conceived and designed the experiments, performed the experiments, analyzed the data, prepared figures and/or tables, authored or reviewed drafts of the paper, and approved the final draft.
- Jared Wray conceived and designed the experiments, performed the experiments, authored or reviewed drafts of the paper, and approved the final draft.
- Cédric Malandain and Gary Barker conceived and designed the experiments, authored or reviewed drafts of the paper, and approved the final draft.

### Data Availability

All software scripts and databases are available at GitHub: https://github.com/gvMicroarctic/PhyloPrimer.

### Supplemental Information

Supplemental information for this article can be found online at http://dx.doi.org/10.7717/peerj.11120#supplemental-information.

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
