# Peer review of "PhyloPrimer: a taxon-specific oligonucleotide design platform"

_PeerJ, doi:10.7717/peerj.11120_

## Round 0.1 · original submission · Minor Revisions

It is requested to address the comments and concerns kindly provided by the two reviewers.

·

Basic reporting

This article introduces a web-based tool for the design of taxon-specific primers and probes for microbial genes, and demonstrates its use by designing and evaluating primers for the genus level and species level detection of Streptococcus in several commercially available mock community DNA preparations.
The writing is excellent and I have only a few suggested changes:
1. Section 2, Implementation - the URL is needlessly given twice.
2. Line 213 - I noticed disagreement in verb tense. Check throughout to make sure tenses used are consistent.
3. Line 265 - Revise to "This is achieved by assigning..."
4. Line 357 - Change "nt" to nucleotide.
5. Line 364 - "The databases are not interchangeable." I doubt this is really what is meant. Maybe a better wording is, "The databases cannot be substituted."

Experimental design

The truly experimental part of this paper is the evaluation of the primers generated by the program. The methods are clearly presented and the results support the conclusions in the discussion section.

Currently section 2. Implementation stands by itself and is not part of the more traditional outline of Introduction, Materials & Methods, Results, Discussion. Because it goes into such detail about the design and workings of the program that it reads something like a methods section, and I wonder if it might better be a part of Materials and Methods.

Validity of the findings

No comment.

Additional comments

No comment.

Reviewer 2 ·

Basic reporting

The manuscript entitled “PhyloPrimer: a taxon-specific oligonucleotide design platform” submitted by Varliero et al. contains the description of interesting and useful bioinformatics tool for designing PCR primers for close-related bacterial species. However, I have some questions and technical recommendations to authors for enhancing their web tool. So, the main list of questions and recommendations using the numeration of strings are the following:
1. String 39: the word “species- specific primers” should be without space after the hyphen, like “species-specific”;
2. String 45: perhaps, should be noted that not only DNA, but also RNA can be used as the template for subsequent amplification (RT-PCR). The authors also can clarify that for PCR diagnostics of bacterial pathogens the most primers detect conservative regions of housekeeping genes;
3. Strings from 87 to 101: the authors reviewed only well-known and popular web tools for primer design, but little information about tools for the primer dimer prediction (e.g. PrimerROC by Johnson et al., 2019 that can accessed at www.primer-dimer.com). The authors also should to cite some tools for PCR primer design like MPrimer (Shen et al., 2010), PrimerDesign-M (Yoon and Leitner, 2015), MRPrimerW (Kim et al., 2016), MRPrimerV (Kim et al., 2016), Oli2go (Hendling et al., 2018), and Uniqprimer (Karim et al., 2019). The pipeline of these tools very similar to PhyloPrimer at several stages (e.g. database screening, primer dimer check and scoring of final oligos). Please, briefly review these tools with similar workflow in the Introduction and make a comparison table of functions and useful options for these tools in the Conclusions;
4. Strings from 109 to 111. Moreover, there are some tools for isothermal amplification methods (e.g. LAMP), which are also suggests annealing sites for oligos of 20 nucleotides. Therefore, these tools also can be used for PCR primer design. These tools include PrimerExplorer (Tomita et al., 2008; http://primerexplorer.jp/e) for primer design and MorphoCatcher (Shirshikov et al., 2019; http://morphocatcher.ru) for taxon-specific target screening using similar approach described for PhyloPrimer. I very surprised that authors use the same target DNA screening approach with alignment of orthologous genes and masking taxon-specific nucleotide positions without reference to the MorphoCatcher pipeline. The difference is that PhyloPrimer allows to automatize some steps of searching the orthologous sequences, making an alignment, but these points should be reviewed in your manuscript;
5. String 145. The web address can be shown with http:// or https:// protocol. After the web link the space was incorrectly used;
6. String 148. The time range can be shown as “40–50 minutes” and “5–10 minutes” (where “–“ is N-dash);
7. String 151. The web links are without type of protocols;
8. String 152. The web link is without type of protocol;
9. Strings from 194 to 198. This part of the PhyloPrimer pipeline is identical with the MorphoCatcher pipeline. Please, clarify the difference of your approach;
10. Strings from 203 to 204. Please, clarify how your pipeline can increase the level of taxon specificity using the certain nucleotide sequence. Some housekeeping genes as single sequence are rarely can be used for taxonomy of bacterial strains. In my opinion, the database should be validated by modern taxonomy and systematics, but if the approach uses the GenBank annotation only you should verify all orthologous sequences manually. Thus, automatization of this step should be verified by user or using different overall genome relatedness indexes (e.g. ANI, dDDH, or MLSA). The same question is to the “2.5. Databases” part of the manuscript;
11. Strings from 406 to 407. Please, provide the original non-cropped images of your gels to verify the absence of primer dimers at weights lower than 100 kb;
12. String 428. The “non-specific band” which was assigned to S. mitis or S. pneumoniae should be sequenced using Sanger method. If the sequence flanked by PCR primers for these two bacterial species are identical, the identity should be shown in text or additional figure.

Experimental design

Thus, I have several additional recommendations for enhancing the manuscript and this investigation. The authors should to provide these additional experimental data to clarify practical application of the PhyloPrimer tool:
1. The gels after conventional PCR are good for demonstration size of the amplicons, but if the authors claim that the PhyloPrimer tool can be used for design primers and probes for qPCR (strings 48 and 53) I highly recommend to perform additional qPCR experiments with EvaGreen intercalation dye and melting curve analysis. The melting curve analysis can verify that each DNA band on gels is a single PCR product, because sometimes one DNA band contains two different sequences with the same length. Moreover, the melting curve analysis will be useful for evaluation of primer specificity during amplification;
2. If differentiation of S. mitis and S. pneumoniae is difficult with the rpoB sequence, maybe the authors try to perform design primers to target another housekeeping gene? It is interesting to perform design of primers for the same PCR conditions (e.g. annealing temperature and magnesium ions) to claim the application of the PhyloPrimer tool for designing primers for multiplex PCR;
3. It is highly important for phylogenetic analysis to use input sequences from validated database, but not from GenBank annotations. Modern phylogenomics reveals new species and genera each year from the deposited genomes and often some different species have very similar sequences of housekeeping genes. I recommend to use some mathematical approach for preliminary analysis of your databases, for example ANI (Goris et al., 2007) and core-genome MLSA (Neumann et al., 2019). Such approach enhance the accuracy of your primer design between target and non-target taxons.

Validity of the findings

I think that the PhyloPrimer tool will be useful tool for many researchers in molecular diagnostics. However, the manuscript can be enhanced with additional experimental data of primer specificity using the melting curve analysis. Computational or manual validation of the database used for creation of target and non-target groups is also very important for taxon-specific primer design.

Additional comments

Therefore, I think that after major revision of the manuscript this research work can be published in the PeerJ journal. I hope that authors can finish these additional investigations soon.

Best regards and stay healthy,
Your anonymous reviewer

---

## Round 0.2 · accepted · Accept

Thank you for the revision and for satisfactorily addressing the reviewers' comments.

·

Basic reporting

The authors have adequately answered my previous comments for this section.

Experimental design

The authors have adequately answered my previous comments for this section.

Validity of the findings

No comment.

Additional comments

No comment.

Reviewer 2 ·

Basic reporting

No comment.

Experimental design

No comment.

Validity of the findings

No comment.

Additional comments

So, the authors made a user-friendly web-based tool for primer design. I only want to note that the References contain some incorrect formatting of Latin names for microorganisms, names of genes, incorrect symbols (e.g. "-" instead of M-dash), cutted titles of articles, and in some cases incorrect format of author's name. I agree that this tool will be helpful for many researchers, therefore the article can be accepted.

Annotated reviews are not available for download in order to protect the identity of reviewers who chose to remain anonymous.